# Integrating the Roles of Midbrain Dopamine Circuits in Behavior and Neuropsychiatric Disease

**DOI:** 10.3390/biomedicines9060647

**Published:** 2021-06-07

**Authors:** Allen PF Chen, Lu Chen, Thomas A. Kim, Qiaojie Xiong

**Affiliations:** 1Department of Neurobiology and Behavior, Renaissance School of Medicine at Stony Brook University, Stony Brook, NY 11794, USA; allen.chen@stonybrookmedicine.edu (A.P.C.); lu.chen.2@stonybrook.edu (L.C.); thomas.kim@stonybrookmedicine.edu (T.A.K.); 2Medical Scientist Training Program, Renaissance School of Medicine at Stony Brook University, Stony Brook, NY 11794, USA

**Keywords:** dopamine, dorsal striatum, ventral striatum, substantia nigra pars compacta, ventral tegmental area, neural circuits, auditory striatum, dorsomedial striatum, dorsolateral striatum, nucleus accumbens, hippocampus, Parkinson’s disease, addiction, Alzheimer’s disease, schizophrenia, D1 Receptor, D2 Receptor, dopamine transporter

## Abstract

Dopamine (DA) is a behaviorally and clinically diverse neuromodulator that controls CNS function. DA plays major roles in many behaviors including locomotion, learning, habit formation, perception, and memory processing. Reflecting this, DA dysregulation produces a wide variety of cognitive symptoms seen in neuropsychiatric diseases such as Parkinson’s, Schizophrenia, addiction, and Alzheimer’s disease. Here, we review recent advances in the DA systems neuroscience field and explore the advancing hypothesis that DA’s behavioral function is linked to disease deficits in a neural circuit-dependent manner. We survey different brain areas including the basal ganglia’s dorsomedial/dorsolateral striatum, the ventral striatum, the auditory striatum, and the hippocampus in rodent models. Each of these regions have different reported functions and, correspondingly, DA’s reflecting role in each of these regions also has support for being different. We then focus on DA dysregulation states in Parkinson’s disease, addiction, and Alzheimer’s Disease, emphasizing how these afflictions are linked to different DA pathways. We draw upon ideas such as selective vulnerability and region-dependent physiology. These bodies of work suggest that different channels of DA may be dysregulated in different sets of disease. While these are great advances, the fine and definitive segregation of such pathways in behavior and disease remains to be seen. Future studies will be required to define DA’s necessity and contribution to the functional plasticity of different striatal regions.

## 1. Introduction

A central goal in neuroscience is to understand the basic functions of the brain and how these functions may go awry in neuropsychiatric and behavioral disorders. In successful cases, there is a bidirectional relationship in which basic studies inform and learn from pathological phenomena. An emerging exemplar of this is dopamine (DA), a major neurotransmitter released throughout the brain that has drawn widespread interest from basic and clinical neurosciences. DA neurons degenerate and a lack of striatal DA is linked to the movement deficits seen in Parkinson’s disease, a major neurodegenerative disorder. Indeed, clinical treatments and rodent models of recovery draw upon DA repletion in the CNS [1,2]. These observations form the basis of major neuroscience curricula and how the basal ganglia contribute to movement. Another major success story in cognitive neuroscience is the finding that DA acts as a reward prediction error (RPE) signal and contributes to reinforcement learning. Linked to major brain functions in movement and reward processing, it is unsurprising that many neuropsychiatric diseases arise from DA dysregulation. The goal of this review is to dissect the neural circuit-dependent roles of DA in behavior and attempt to link these functions to DA’s dysregulation in different brain diseases. 

We attempt to draw insight from what we can learn from both basic and pathological perspectives on DA. While there have been major advances, many lines of DA investigation in health and disease have remained largely separate. It is critical to cross fields and start asking questions such as “Do models of Parkinson’s disease rely on DA dysregulation across the entire striatum or specific subregions? What roles do striatal subdivisions have in addiction pathogenesis? Or schizophrenia?”. When considering the cognitive and perceptual deficits of Parkinson’s disease and other DA dysregulation disorders, it is tantalizing to speculate on the nuances of DA’s function. We adopt a current view in the literature that there may be separate channels through which DA functions, as determined by specific anatomical projections [3,4,5]. While we provide some evidence that supports this hypothesis, it should be noted that it is still an open question whether a single or single set of principles underlie DA’s function in multiple behaviors. Nevertheless, we systematically discuss the basic function of major neural circuits regulated by DA. Specifically, we dissect circuits in the dorsal striatum, the ventral striatum, and the hippocampus. We attempt to be comprehensive in this regard by delineating general function, Dopamine Receptor-1 (D1R) vs. Dopamine Receptor-2 (D2R) medium spiny neuron pathway differences, and DA modulation therein. We note along the way the importance of the D1R/D2R dichotomy, as this relates to theoretical and clinical aspects of using dopaminergic pharmacological agents. Within this paradigm, it is widely appreciated that DA tends to increase the activity of D1R-expressing neurons and decrease the activity of D2R-expressing neurons [6,7,8,9]. We discuss how this physiological mechanism extends to behavioral phenomena. Then, we discuss how major brain diseases are related to this circuit-function framework. As the literature regarding DA in behavior and disease is expansive, we primarily focused on recent advances in DA regulation and do not cover the remarkable literature regarding the role of non-DA cell types in the regulation of DA output. For these relevant studies, please see [10,11].

## 2. Behavior to Neuroanatomical Circuit-Function

### 2.1. Anatomical Overview

It has long been appreciated that DA transmission occurs broadly throughout the brain and exerts effects on an arsenal of behaviors. Canonical organization of dopamine-containing nuclei was established in the 1960s and 1970s with the development of the then introduced formaldehyde histochemical techniques [12]. Carlsson and colleagues took advantage of the chemical interaction of catecholamines and formaldehyde, which produces fluorescent by-products [13,14]. Later, the development of immunohistochemical methods for the detection of catecholamine-synthesizing enzymes allowed for the elaboration of all catecholaminergic nuclei [15,16]. Critically, this work determined that the substantia nigra pars compacta (SNpc) and the ventral tegmental area (VTA) are major midbrain complexes that are dopaminergic in nature. A general picture arises in which dopaminergic neurons are densely populated in the VTA and more so in the SNpc [11,17,18]. The SNpc primarily projects to the dorsal striatum (DS; caudate/putamen in primates) while the VTA primarily projects to the ventral striatum (VS)/nucleus accumbens (NAc). Previous studies have created a dichotomous understanding of the DA nuclei that implicates the SNpc’s nigrostriatal pathway in movement and the VTA’s mesolimbic pathway in reward processing and motivation [19,20,21,22]. This dichotomy, while oversimplified considering recent studies (see [3,23,24,25]), has been useful to understand neural and behavioral dysfunction in disease. We will discuss the basic function of the medial and lateral DS subdivision (DMS and DLS) and relevant studies on the role of DA modulation shown here. Similarly, we will discuss parallel roles in behavior for the VS. 

Along the way, we also discuss the respective functions of the D1R and D2R genetically defined anatomical pathways. We note here that the D1R and D2R receptors are part of the D1-like family (including D1R and D5R) and D2-like family (including D2R, D3R, and D4R), respectively [7,8,26]. For the sake of focusing on behavioral functions of DA, we have limited our discussion to the D1R and D2R pathways, which are the primary constituents of both the dorsal and ventral striatum [27,28]. However, it is important to note that the other receptors (D3-5) are pharmacologically, behaviorally, and clinically relevant. For instance, the D3R is expressed in and regulates the prefrontal cortex, with implications in modulating memory function in schizophrenia [26,29]. While not delineated in this review, we encourage the consideration of all DA receptor subtypes in order to adopt a complete framework of DA in behavior and disease. After discussing the major striatal subregions, we lastly add in research performed in the hippocampal field regarding DA. In Figure 1, we summarize the different behavioral functions related to the different parts of the DS, as well as the different roles of DA associated with the striatum in these regions. In Table 1, we summarize the key points from our review of recent research. We note here that these ongoing observations and definitive conclusions about different striatal subregions may require further systematic study. 

### 2.2. SNpc and the Dorsal Striatum

The bulk of the DS projects to the SNpc and receives projections back from SNpc DA neurons [19,21]. Overall, the DS has been divided into two major behavioral pathways on the basis of molecular and anatomical evidence, the D1 “direct” and D2 “indirect” pathways. In addition, the DS has traditionally been divided into two behaviorally distinct regions, the DMS and the DLS. The function of these two structures have been attributed to goal-directed behaviors and habit-based behaviors, respectively [30,31,32]. A related categorization, the DMS is denoted as the associative striatum, while the DLS is the sensorimotor striatum. There is a parallel anatomy that suggests that the DMS and DLS receive differential cortical inputs, with the DMS receiving more limbic inputs and the DLS receiving more motor inputs [33,34]. We could consider that, if the DMS and DLS serve different functions, it is perhaps the case that their respective DA inputs also serve different functions.

#### 2.2.1. DMS Circuits in Goal-Directed Behavior and Beyond

*DMS General Function*: The DMS is a striatal subregion important for goal-directed behaviors. Observed in a variety of reward-based behaviors, as animals first learn a task to obtain reward, they tend to choose a goal-oriented rather than a stimulus-oriented strategy. For early lever pressing or navigational tasks, this means that the animal interacts with the environment more flexibly in order to obtain a reward. In early studies, groups found that, after lesioning the DMS, rats would perform water maze tasks in a more cue-responsive manner than a goal directed one [35,36]. The opposite was the case for animals with DLS lesions. This divide between the DMS/DLS was further supported by in vivo recordings that demonstrate that DMS activity is greater earlier on during instrumental training, whereas DLS activity is more prominent after extended training. Thus, the current view is that the DMS is important for goal-directed behaviors whereas the DLS is more important for habit-based or skill-honing behaviors.

In line with a role in goal-oriented behavior, DMS neuronal activity correlates with flexible switches in task contingencies in rats that perform a dynamic go/no-go task [37]. In another task dimension, it has been demonstrated that the DLS performs action generalization whereas the DMS is better suited for action discrimination and differential utilization. This study suggests that the DMS and DLS may behaviorally compete with one another for adopting particularly motor strategies when engaging in the environment. Adding to its role in flexible action selection, the DMS is also implicated in perceptual actions and sensory encoding. This poises the DMS as a structure to encode actions within the perceptual contexts of the environment, which is related to previous studies that implicated the DMS in navigational and water maze tasks.

The DMS indeed receives visual, somatosensory, and some sources of auditory input [33,34]. Interestingly, the DMS functionally responds to both somatosensory and visual input, whereas the DLS primarily responds to somatosensory inputs [38]. In an interesting brain-machine interface study, mice were trained to control visual cortical activity in order to generate favorable online task outcomes [39]. They found that inactivation of the DMS impairs the ability of mice to learn this visual cortical control task. This study allows us to speculate whether visual corticostriatal projections or other parts of this basal ganglia segment may allow for special types of visual learning. Furthermore, activation of DMS striatal neurons during a perceptual decision-making task causes contralateral bias. Importantly, in this task, the mode of activation occurred only during stimulus onset, suggesting a role for the visual stimulus encoding in the DMS. In support of this, a recent systematic study utilized neuropixel probes and widefield calcium imaging to simultaneous probe the activity of cortical and striatal activity across the medial–lateral axis. During a visual action association task, they found that DMS neurons responded to visual stimuli in a manner consistent with cortical input activity, whereas DLS neurons responded more to rewarded sensation or licking action. This study dynamically demonstrated distributed perceptual responses and is consistent with previous studies.

*D1* vs. *D2 in DMS:* Many studies that provide evidence for the canonical divide between the direct and indirect pathway have been carried out in the dorsomedial striatum [40]. Mechanisms of DA strongly support the behavioral divide in that DA tends to excite or enhance synaptic transmission in D1R MSNs and depress activity in D2R MSNs [6,8]. This divide has been investigated using the early adoption of optogenetic tools. These early experiments indicate that D1R MSNs reinforce lever pressing behavior whereas D2R MSN stimulation serves to signal transient punishment for the animal. Consistent with this, Balliene’s group recently demonstrated that learning-associated plasticity has opposite signs for the two pathways in the DMS, with synaptic enhancement of D1R MSNs and depression in D2R MSNs [41]. Other studies in value-based decision making have demonstrated that both D1R and D2R’s MSNs play a role in decisions. In line with this, it was previously demonstrated that the DMS direct pathway is involved in action selection [42]. Here, optogenetic stimulation of the D1R MSN population induces contraversive bias, whereas the D2R population induces ipsiversive bias. It is important to note that the above two studies dissect different behaviors, with the former focusing on outcome evaluation and the latter on ongoing decision output structure. Thus, regarding the direct vs. indirect pathway dichotomy, both appear to play opposing roles in different forms of goal-directed decision-making and learning. While more work will be required to dissect anatomical differences in this opponency, it explains different aspects of dopaminergic antagonists such as loss of motivation and deficits in learning.

*DA in Dorsomedial Striatum*: DA release in the DMS has been found to correspond to reward, movement, and aversive behavioral cues. Having discussed the behavioral divide between the DMS and DLS, several recent studies have attempted to address differences in DA signaling in these two regions. For instance, Lerner et al. studied the SNpc-DMS and SNpc-DLS circuits as being separate both in anatomy and function [18]. Using whole brain mapping with rabies viral targeting and brain clearing strategies, they found that DLS-projecting SNpc neurons receive both DLS and DMS inputs but have a bias for DLS input. The converse is true for DMS-projecting SNpc DA neurons. This suggests that the SNpc DA neurons projecting to the DMS or DLS are differentially regulated by reciprocal GABAergic MSN inputs. In agreement with the prior literature, they also demonstrated that the DLS-projection SNpc neurons are located more laterally in the SNpc, while the DMS-projected neurons are located more medially. Interestingly, they found that other subcortical and cortical areas appear to project to these compartments of the SNpc somewhat equivalently. Consistent with work demonstrating the role of DMS in action selection, DA inputs to this region appear to encode contralateral actions in contrast with reward prediction error encoding demonstrated by the ventral striatal DA. Interestingly, in a recent preprint study by Lerner’s group, DA responses in the DMS were found to correspond to compulsive behaviors. They showed that task-relevant increases in DMS DA corresponded to which individual animal would express compulsive behavior. Optogenetic stimulation of DA terminals in the DMS increased compulsive but not habitual types of behavior. This has implications for our understanding of DA pharmacological agents in OCD and attention disorders [32]. While there is a gradient of encoding a variety of variables amongst DA neurons [3], it appears that DMS DA may correlate well with what the striatal target neurons seem to perform in terms of goal directed behavior.

#### 2.2.2. DLS Circuits—Habit Formation and Beyond

*DLS General Function*: A large body of literature suggests that the DLS is involved in sensorimotor encoding and habit formation. Appropriately, in comparison to the DMS, the DLS receives more input from the primary somatosensory and primary motor cortices [33,34]. In contrast with the DMS, DLS striatal neurons appear to have higher activity in rodents that have undergone extended bouts of training, and thus appear to be more correlated with skill honing and habit execution. In line with this, a recent study showed elegantly that the DLS is increasingly modulated by DA as an animal becomes an expert at an instrumental task. However, just as there is regional topographic segregation in the motor cortex, one can ask whether this applies to the DMS/DLS subdivisions for action control. A recent study topographically mapped out the different pathways in basal ganglia, comparing the DMS, the DLS, and the ventrolateral dorsal striatum [43]. This group showed that the DMS pathway is important for contraversive turning whereas the ventrolateral dorsal striatum is important for contraversive licking. They argue that, while previous studies have implicated the DLS in both contralateral licking and turning, other regions of the DS more potently control these specific actions. This is overall consistent with the literature that suggests that basal ganglia circuits are important for contralateral actions. It will be important to consider how these data fit in the context of goal-directed vs. habitual dichotomies. Further, it will be important to understand how sensory input into these different regions may map onto specific behaviors. For instance, given the preponderance of S1 inputs in the lateral DS, one would predict that whisker stimulation best triggers licking behaviors in a habitual manner.

Previous studies show the importance of the dorsolateral striatum in simple discrimination tasks [44]. In addition, the DLS has also been implicated in perceptual-based behaviors that require somatosensation. Whisker-based choices recruit the DLS circuit and appear to be important for the timing of actions initiated by the DLS [45]. In addition, the DLS striatal neurons seem particularly to respond to somatosensory stimulation, whereas DMS striatal neurons also respond to visual stimuli [38]. Lastly, the DLS striatonigral pathway is involved in encoding whisker stimulation responses in a Go/No-go paradigm. In light of these studies, the DLS is poised to perform basic sensorimotor integration with strong communication with motor and somatosensory cortices (Figure 1).

*D1* vs. *D2*: As with the DMS, the DLS also consists primarily of MSNs of the direct and indirect pathways. Compared with the DMS, the evidence for segregated D1 and D2 pathways in the DLS is less abundant. Recordings in the central part of the DS are consistent with this, demonstrating that both D1 and D2 pathways are concurrently active during movement [46]. A newer emerging model postulates that the pathways are intertwined, and both are required for purposeful action execution. Consistent with this, evidence suggests that the D1 and D2 pathways in the DLS both act to reinforce optogenetically linked lever press actions. However, the action strategies supported by the D1 and D2 pathways are different. In tangent with this study, other groups have also shown nuanced learning roles for the D1 and D2 pathways in the DLS [47,48]. These studies indicate that the D1R pathway in the DLS is critical for initial action contingency adoption, while the D2R pathway may play more of a role in sculpting and refining behavioral strategies. In the context of habit formation and expression, an imaging study conducted indicates that both pathways play time-specific roles in habit expression [49]. Interestingly, in this study, activation of the D2R pathway did not predict habit suppression, and both pathways underwent strengthened plasticity during habit formation. In relation to sensorimotor behaviors, Sippy et al. demonstrated that D1R pathway activity correlates with whisker sensory responses and that optogenetic inhibition of this pathway reduces conditioned licking responses. In conclusion, both D1R/D2R pathways in the DLS appear to be important for learning and habits but to different extents.

*DA in Dorsolateral Striatum*: DA in the DLS has been documented to correlate with sensory, motor, and learning variables. This is similar but distinct from the variables seen in the DMS. As lesions of the DLS lead to impairment of habit formation, removing DA input to this region via neurotoxin delivery also results in impaired habit formation [50,51]. In comparison to the DMS, Lerner et al. found the DLS DA dynamics appear to have similar reward processing but a greater response to aversive events. A recent study showed elegantly that the DLS becomes increasingly modulated by DA as an animal becomes an expert at an instrumental task. In an interesting set of studies, Palmiter’s group showed that, in dopamine deficient mice, rescuing DA only in the DLS is sufficient to preserve the ability to learn cognitive tasks, albeit with a slower rate of learning [52]. This is important because it indicates that DA processing in the DLS may contribute to aspects of learning and motivation, which are attributes primarily associated with the DMS or VS. It should be noted that the mice generated in these studies have been found to have severe deficits in learning and motivation that are comparable to those in mice with bilateral OHDA depletion models. Corresponding to the previously discussed sensorimotor role of the DLS, Ketzef et al. found that DA depletion in the DLS impairs bilateral somatosensory processing in MSNs [53]. This work may reflect some of the sensory deficits seen in Parkinson’s disease.

#### 2.2.3. Auditory Striatum

*General Function*: Early on in the striatal literature, LeDoux’s group characterized tonal responses in the striatum of anesthetized cats. Since then, a variety of groups have probed the sensory representation in the striatum across visual, auditory, and somatosensory modalities. The auditory cortex projects strongly, particularly to the tail or caudal extreme of the DS. These projections are not defined or experimentally investigated by studies on the more rostral DMS and DLS. Based on its strong auditory cortical input, here, we synonymously refer to the “tail” or “caudal” striatum (as denoted in previous literature) as the auditory striatum. We note here that we do not discuss the auditory striatum in non-rodent species; there is anatomical and behavioral evidence for an auditory striatum correlates in nonhuman primate (extensively reviewed here). There are several anatomical characteristics of the auditory striatum that distinguish it from the DMS and DLS (Figure 1). Anatomically, the auditory striatum receives projections primarily from sensory cortices and thalami [33,34]. Additionally, it is innervated by the lateral region of the SNpc or the substantia nigra pars lateralis which receives efferent projections that differs from the rest of the SNpc. Furthermore, this region has a subregion that is particularly rich in D1R MSNs (adjacent to the globus pallidus externus), unlike the rest of the dorsal striatum, which has an approximately even distribution of D1R and D2R MSNs [54]. Taken together, the auditory striatum is anatomically poised to perform functions that differ from the DMS/DLS by specifically integrating sensory inputs. However, as reviewed in the above sections, there is electrophysiological and in vivo evidence for strong visual and somatosensory processing in the DMS and DLS, respectively.

Zador’s group found that auditory corticostriatal projections functionally control arbitrary sound–action decisions in rats. This study is important because it depicts a potential pathway through which arbitrary sound representation in the auditory cortex can be translated into actions through the basal ganglia. This serves as a general mechanism through which the basal ganglia may contribute to perceptual decision making. Furthermore, we found that the potentiation of auditory corticostriatal synapses underlies the acquisition of sound–action learning [55]. This depicts a neural mechanism through which sound–action learning can occur through specific corticostriatal interactions. This is consistent with DA theory in how DA modulates learning in the striatum by potentiating corticostriatal synapses. Beyond cortical connections, we have also shown that the auditory thalamus functionally regulates the region of the auditory striatum [56]. These thalamic connections appear to be important for controlling the gain of sound representation within the auditory striatum. Similar to corticostriatal or thalamostriatal intervention, optogenetic activation of D1R MSNs in the auditory striatum demonstrates that it can bias auditory decisions in a manner that is consistent with tonal stimuli but not simple contraversive choice selection [57]. Thus, the striatal MSNs in the region are important for stimulus encoding and decision-making enforcement. However, how this region fits into the VS to DLS continuum is unclear and it may potentially be the case that this region has specialized sensory functions. More studies examining the auditory striatum in different behavioral paradigms are needed to draw critical comparisons.

*D1R* vs. *D2R in the Auditory Striatum:* Like its general role in the basal ganglia, the auditory striatum’s direct and indirect pathways remain to be clarified. As mentioned previously, Jaramillo’s group optogenetically manipulated the D1R pathway and demonstrated a role for auditory-guided decisions that was distinct from a general movement bias demonstrated in other regions of the dorsal striatum [57]. A recent paper from Zhang’s group characterized the role of the auditory striatum in defensive behaviors [58]. They demonstrated that auditory corticostriatal projections targeting D2R MSNs are particularly important for freezing behavior in response to fear-inducing auditory cues. With these two studies combined, it appears that the function of D1R/D2R MSNs may support traditional Go/No-go basal ganglia models specifically in auditory or sensory-dependent behavioral contexts. Further studies will be required to fully characterize the function and input/output dynamics of these pathways.

*DA in the Auditory Striatum*: It has been shown that the posterior extreme of the striatum receives differential cortical, thalamic, and dopaminergic input [33,34,59]. In particular, the SNpc DA neurons that project to the auditory striatum appear to reside in the lateral most part of the SNpc (denoted as the substania nigra pars lateralis, SNL), and receive a different set of inputs in comparison to other SNpc/VTA DA neurons. In addition, these neurons appear to respond to novel, intense, and aversive stimuli [60,61]. Menegas et al. demonstrated that these neurons do not appear to signal a traditional RPE that has been found in the SNpc and VTA neurons that presumably project to the more canonical DS and VS. DA antagonism and OHDA depletion of DA in the auditory striatum reduces an animal’s ability to learn about threatening stimuli. In this study, they used air puffs and bitter taste to signal aversive stimuli. In comparison with DA signals in other striatal regions, auditory striatal DA release seems to be the most distinct. Future studies will be important for definitively demarcating this region and determining whether it is sufficient to denote this overall region as the “tail striatum” or whether it is based on specific sensory inputs. Furthermore, it will be important to investigate whether the auditory striatum’s DA input will be important for other types of sensory learning and decision making.

### 2.3. VTA and the Ventral Striatum

*Ventral Striatum—Motivation and learning*: Having discussed the differences in the lateral and medial subdivision of the dorsal striatum, it is important to consider the foremost subdivision of the striatum across the dorsal-ventral axis. In comparison with the DMS, the VS or nucleus accumbens is thought to be more related to emotional, social, and reward processing. Anatomically the ventral striatum has been demarcated as consisting of the ventral and medial parts of the dorsal striatum, the nucleus accumbens, as well as the olfactory tubercle system. Here, we primarily consider the nucleus accumbens, which is subdivided into core and shell components, on the basis of immunocytochemical architecture, as well as input–output connectivity [11]. Excitotoxic lesions of the nucleus accumbens core impact the learning of task acquisitions [62]. Akin to lesions in the DMS previously discussed, the core lesions’ effect on stimulus–response learning mainly impacts early learning and had little effect if rats had already learned the task. In comparison to the core’s role in reinforcement learning, the shell appears to play more of a role in motivated behavior and the locomotion of conditioned response expression [63]. Overall, the nucleus accumbens core and shell appear to play dual roles in motivation and learning. This divide, while not complete, appears to be analogous to the DMS and DLS divide previously discussed.

*D1* vs. *D2*: Both the core and shell of the nucleus accumbens are comprised of D1R and D2R MSNs. However, in comparison to the dorsal striatum, these pathways project to different but analogous regions. In canonical contexts, the D1R MSNs project to the VTA while the D2R MSNs project to the ventral pallidum. Paralleling the neuroanatomical dichotomy, studies suggest that the D1R pathway encodes positive valence while the D2R pathway encodes negative valence [28]. However, a major study by Kupchik and colleagues demonstrated that the D1R and D2R pathways are not label-lined, and thus genetic or pharmacological interventions of the respective receptors should be interpreted with caution [64]. Another complicating factor is that there are populations of MSNs that co-express D1R and D2R [65]. Thus, as with the DLS and DMS, the evidence for a D1R/D2R functional and behavioral divide is apparent but complex. In comparison to the DLS/DMS, where influential models of Parkinson’s disease support a D1R/D2R opponency, the divide is less clarified in the VS.

*DA in Ventral Striatum*: In comparison with the DMS and DLS, DA release in the VS appears to correspond more directly with reward processing and hedonic value. However, this divide is not clear cut and it is important to consider recent studies. The VTA has major projections to the frontal cortex and the ventral striatum (VS) in the respective mesocorticolimbic and mesolimbic systems. Traditionally, these associations have led to the canon conclusion that, in comparison to the SNpc’s role in movement and ongoing behavioral modulation, the VTA is more involved in emotion, learning, and motivational processing. It is important to note here that this divide is neither distinct nor absolute [5,23,24,25]. Advancing the SNpc’s role in movement, recent and previous studies have shown that nigrostriatal activity can correspond to transiently defined movements [23,66,67,68]. However, supporting a role for the VTA system in movement, early studies have shown that DA depletion or antagonism of the VS results in hypoactivity in rodents. Conversely, DA agonism in the VS causes an increase in spontaneous movements. Consistent with this, recent studies have reinstated the importance of movement and kinematic modulation in VTA moment-to-moment responses [3,69,70]. Hughes et al. recently discovered quantitative and causal relationships between VTA activity and movement impulse vectors. Importantly, they showed that VTA DA neuronal activity correspond with different types of movement, regardless of reward or aversive contexts. While these more recent experiments consider the dissociation or confounding of movement with general motivation, it is still theoretically difficult to separate these variables, especially in the context of learning. Thus, further studies will need to identify how the VTA modulates movement variables and how this mode of DA release contributes to motivated behaviors. Additionally, it will be important to consider how DA instantaneous modulates downstream cellular targets in order to enforce movement changes.

From both anterograde and retrograde tracing studies, it has been identified that the lateral portion of the VTA generally projects to the core of the nucleus accumbens, whereas the medial portion projects to the shell [47,69]. This has been found to be significant considering the large body of literature that distinguishes these two accumbal regions in terms of their capacity to process reward, aversive stimuli, and motivation. It appears that VTA DA neurons projecting to the medial nucleus accumbens are involved in aversive fear conditioning, as controlled by the lateral hypothalamus.

Beier et al. systematically characterized the input–output relationships of the VTA. They found that the VTA population projecting to the medial nucleus accumbens is a distinct population of DA neurons that is largely non-overlapping with the VTA population projecting to the lateral nucleus accumbens. Like other studies, they also found that these populations had different roles in motivating reward behavior. A recent study found that the lateral VTA projections to the NAcc core is specifically involved in generating ongoing motivational behavior [71]. They found that, in general, the VTA DA population encodes reward prediction error, the signal attributed to reinforcement learning. Conceptually, however, such RPE learning signals would not be attributed to a role in enforcing ongoing behavior, as such behaviors depend on already-learned contingencies. Thus, the researchers hypothesized that separate neuronal mechanisms must be involved in order to dissociate signals for learning and motivation. Moreover, they hypothesized that both could be attributed to the mesolimbic pathway. They found that DA levels in the NAcc specifically fluctuated with ongoing states of motivation, whereas the levels in other regions of the NAcc only appeared to change with RPE signaling. Thus, this study points towards the differences in function of different VTA DA projections.

### 2.4. VTA and the Hippocampus

Thus far, we have discussed the striatum and its function as a major DA afferent recipient. While these projections represent major circuit pathways through which DA modulates behavioral output, DA nuclei also provide input to major cortical and hippocampal regions. Here, we limit our discussion to the role of DA in hippocampal circuits, as subsequent sections will highlight a role for this modulation in health and disease. Early evidence utilizing electrolytic lesions of dopaminergic nuclei indicates that the hippocampal formation receives input from both the VTA and SNpc [72]. In addition, circuit plasticity of the hippocampal formation was seen to be modulated by DA pharmacological agents [73]. In a variety of studies, D1R receptors have been found to be expressed in the molecular layer of the dentate gyrus, while D2R receptors have been found to be expressed in the hilus. Thus, at the anatomical and histochemical level, evidence points towards a role for DA in hippocampal function.

Influential theories about how the VTA and the hippocampus are involved in a novelty–reward loop plays into how memories are formed. Briefly, hippocampal connections to the VTA trigger firing in response to novel information. This increased VTA firing is proposed to increase DA in the hippocampus and therefore support the synaptic plasticity underlying different types of memory formation. The optogenetic activation of VTA DA cells have been shown to trigger CA1 replay activity and improve spatial memory. Consistent with this, VTA DA neuron activation can bidirectionally modulate Schaffer collateral synapses. Furthermore, one group demonstrated that concurrent activation of CA1 and the medial forebrain bundle was sufficient to create place preference and a memory trace [74]. Other work demonstrates that the source of DA for modulation of hippocampal circuits may originate from the locus coeruleus in addition to the VTA [75,76]. Nevertheless, the role of DA as an important modulator for hippocampal memory function points toward a potential role in cognitive function and dysfunction.

### 2.5. Conclusion to Behavior to Neuroanatomical Circuit-Function

Emerging studies of DA and striatal subregion-dependent function provoke multiple themes. Support arises for fundamental differences between striatal regions including the DMS, the DLS, the VS, and the auditory striatum. The VS and DMS have a greater influence on flexible, goal-directed behaviors, while the DLS is more related to stimulus–action encoding and habitual expression. With caveats, original lesion and DA pharmacological studies have been supported by newer sophisticated neuroanatomical tracing, optogenetics, and behavioral paradigms. While there is support for differences in DA function in each of the subregions, aside from auditory striatal DA, there is a degree of convergence at the physiological level. Thus, the specific function of DA in each region may rely mostly on the integrated processing of corticostriatal, thalamostriatal, and nigrostriatal/mesolimbic inputs. Future studies are needed to clarify these neuromodulatory circuits. A thought-provoking circuit question that arises is how to think about D1R and D2R MSNs in each respective region. We have cited studies that primarily refine or refute the original models in movement and reward learning. One trend that needs to be verified is whether the D1/D2 dichotomies hold more strongly when studying the VS and DMS, whereas breakdown occurs when studying the DLS. It is perhaps the case that the DLS as a sensorimotor region is more strongly influenced by complex ongoing locomotor behaviors that may obscure sensory/learning neural dichotomies during experiments. In conclusion, working models have been and continue to be refined but, to deeply understand DA’s role in behavior, there needs to be further clarification of DA receptor pathways in the different regions of the basal ganglia.

## 3. DA in Neuropsychiatric Disease

*Overview*: As we have reviewed, DA plays major behavioral roles in modulating major brain regions of the CNS. Thus, it is tantalizing to hypothesize that such functions go awry in neuropsychiatric diseases. Indeed, there is strong support for this in theoretical accounts of dopaminergic pharmacological therapies. In these next sections, we highlight neuropsychiatric diseases and afflicted dopaminergic pathways. We emphasize that we adopt a biased DA-centric view in this regard, and we refer readers to other sources in order to adopt a holistic view of these disorders. We discuss literature that highlights the molecular and genetic basis for selective vulnerability of DLS-projecting SNpc neurons. We then discuss the circuits involved in addiction, highlighting the VTA-VS pathway in this disease. We also discuss recent evidence that indicates an amyloidogenic vulnerability of the VTA–hippocampal pathway in Alzheimer’s Disease. Lastly, recent evidence has emerged regarding the dopaminergic hypothesis of schizophrenia and how it relates to rodent neural circuits. We summarize these recent findings in Figure 2, where we illustrate the specific dopaminergic circuits implicated in disease. Overall, this serves to describe how particular DA circuits are co-opted in different neuropsychiatric diseases.

### 3.1. Parkinson’s Disease and Dopamine Depletion

One of the most well-known neurological disorders caused by dopamine dysregulation is Parkinson’s Disease (PD). PD is the second largest cause of neurodegeneration [77], which is marked by dopaminergic neuron loss in SNpc [78]. The degeneration of SNpc dopaminergic neurons is thought to be the likely cause the dysfunction of their projection target—the dorsal striatum [79]. Since the dorsal striatum (DS) is responsible for action selection and movement control [46,80,81,82], dysfunction of the DS will lead to movement deficits, such as rigidity, tremor, and shuffling gaits [27,83]. These movement difficulties are the first-found symptoms [84] and are hallmarks in PD cases, serving as evidence for a clinical PD diagnosis [85].

The mechanisms of SNpc neuronal death in PD are a topic of current investigation and debate. Alpha-synuclein protienopathy and mitochondrial dysfunction are two major mechanisms that have been highly investigated. Given its molecular and clinical precedence, alpha-synuclein pathology has been believed to be one of the most probable causes of PD [86,87]. Alpha-synuclein proteins are expressed by neurons across different brain regions and interact with pre-synaptic dopaminergic machinery [88,89,90], such as the dopamine transporter [91]. In Parkinson’s Disease, α-synuclein are aggregated in the cytoplasm and neurites of SNpc dopaminergic neurons, forming Lewy Bodies and Lewy neurites [92]. These aggregated α-synuclein are toxic to neurons and lead to neuronal cell death molecular cascades. However, early neuronal loss in SNpc has also been shown to precede α-synuclein aggregation. Other than α-synuclein pathology, mitochondrial dysfunction is also believed to be responsible for dopaminergic neuron degeneration. Consistent with this, mutations of mitochondria function-related genes have been found to contribute to familiar PD. However, in sporadic PD, which has a later onset in comparison to genetic causes, it has been difficult to prove whether mitochondrial dysfunction is the original cause of dopaminergic neuron loss or just a result of other pathological events. Whether these two major molecular causes underlie selective vulnerability of SNpc DA neurons has not been investigated.

### 3.2. SNpc DA Neurons’ Selective Vulnerability in PD

As we discussed, motor dysfunction is the hallmark of this neurodegenerative disorder. There were cognitive disorders (e.g., dementia) reported in many of PD cases but there are also PD patients without cognitive abnormalities [93,94]. This phenotype is consistent with the fact that SNpcs degenerate earlier and more severely than the degeneration of VTA in PD brains. In both human PD cases and animal PD models, neurodegeneration follows this conserved pattern in midbrain dopaminergic nuclei [95,96,97]. Dopaminergic neurons in SNpc degenerate more severely than the VTA and within SNpc, and dopaminergic neurons in ventral SNpc degenerate more severely than the dorsal tier of SNr [98,99,100,101]. Different hypotheses have been brought up to explain this selectivity. Differential expression of transcription factors and differences in bioenergetics or metabolism could mark the different vulnerability of SNpc and VTA DA neurons. We discuss several of the major markers of vulnerability below.

Calbindin: One of the markers for selective invulnerability or neurodegeneration-protective is calbindin. Early in 1990, there were papers reporting that, in PD cases, calbindin-(+) SNpc DA neurons were more spared than calbindin-(−) counterparts [102]. Therefore, calbindin has been considered to be a marker of selective invulnerability. The distribution of calbindin-(+) dopamine neurons is mostly concentrated in the VTA and denser in the dorsal tier of SNpc than in the ventral tier [102,103]. This distribution is thus consistent with the vulnerability pattern of PD. Calbindin-(+) dopamine neurons in the dorsal SNpc mainly projects to the medial part of the caudate-putamen, through the rostro-caudal axis [103]. As described in a previous paper, calbindin-(+) dopaminergic projections to the DMS are more resistant to PD pathology, while DLS dopamine projections will degenerate faster [104]. Moreover, the VS receives more calbindin-(+) dopaminergic input from VTA than DS and, therefore, the dorsal striatum suffers from a more severe dopamine deficiency than the ventral striatum in PD. Different vulnerabilities in subregions of the striatum are consistent with the fact that movement deficits are more common and pervasive than other symptoms in PD.

The mechanism of the protective role of calbindin in dopaminergic system has been thought to be related to calbindin’s Ca^2+^ buffering capacity. High supraphysiological concentrations of Ca^2+^ in neurons are generally toxic and such Ca^2+^ could also trigger the aggregation of α-synuclein [105]. This hypothesis is also supported by research indicating that calbindin-(−) dopaminergic neurons (ventral tier of SNpc) have a prominent T-type calcium current and a higher Ca^2+^ amplitude. This is in contrast with the DA neurons of the dorsal tier of SNpc, which only exhibit a small T-type current. Additionally, VTA DA neuron spiking does not depend on calcium channels [98,106,107]. In vitro assays also showed that overexpression of calbindin in cultured dopaminergic neurons could attenuate the death induced by MPP+, a toxin mimicking PD. However, there are arguments about the relationship between calbindin and protection in selective vulnerability, as a paper from 2012 notes by reporting only 2% calbindin-(+) dopaminergic neurons in the dorsal SNpc. The authors argued that the small number might not be enough for neuroprotective function [98,108]. Later, in a macaque study, a much higher number of calbindin+/TH+ co-expression in the dSNpc appeared (34.7%) [109]. This might be due to the different species in the experiments, but this suggests that calbindin might still be able to serve as a selective invulnerability or protective marker.

Electrophysiological property and ion channels: Previous research reported that calbindin shows a subpopulation-specific expression in SNpc DA neurons [106,110]. Therefore, researchers considered calbindin as a marker for different SNpc DA populations. Studies have been focused on the differences of electrophysiological properties of calbindin-(+) and calbindin-(-) DA neurons. Calbindin-(-) DA neurons, which project to DLS, have low-threshold Ca^2+^ depolarization in hyperpolarized conditions, which is absent in calbindin-(+) DA neurons in dorsal SNpc [106]. This low-threshold Ca^2+^ depolarization is accompanied by a large and prolonged Ca^2+^ transient in dendrites of calbindin- DA neurons. As we discussed in the previous section, a high level of Ca^2+^ concentration directly contributes to the vulnerability of ventral SNpc DA neurons. Ca^2+^ could bind to α-synuclein and mediate the interaction of α-synuclein with pre-synaptic sites [105]. A recent paper suggested that calcium promotes the aggregation of α-synuclein [105]. On the other hand, aggregated α-synuclein could bind to calcium pump and further enhance the cytosolic concentration of Ca^2+^ [111]. This positive feedback loop explains why ventral SNpc DAs are at higher risk of α-synuclein aggregation and neurodegeneration. Except for α-synuclein aggregation, higher levels of cytosolic Ca^2+^ also add to the oxidative stress of mitochondria [112,113]. Therefore, ventral SNpc DA neurons have a more vulnerable ER-mitochondrial system, considering the higher levels of cytosol Ca^2+^ concentration. These special electrophysiological properties and higher “daily” Ca^2+^ burdens explain why vSNpc DA neurons degenerate faster and more severely in PD cases, and therefore lead to significant deficits in DLS function.

Studies have also tried to understand the different Ca^2+^ concentrations and different electrophysiological properties of dorsal and ventral SNpc DA neurons through the different expressions and functions of ion channels. Several important ion channels, mostly potassium and calcium channels, have the possibility of being markers of selective vulnerability. Ion channels such as GIRK2 (G-protein-regulated inward-rectifier potassium channel 2), SKs (calcium-activated potassium channels), T-type calcium channels, and L-type calcium channels have been analyzed through immunostaining assays [114,115]. The general idea is that the expression of these ion channels in SNpc subpopulations remains similar, although there are debates regarding some other channels in different research [115,116]. These conflicts are likely due to the affinity or selectivity shortage of immunostaining protocols. Although the expression of channels shows no significant difference, as we mentioned above, the function of T-type calcium channels could serve as a marker of selective vulnerability. Ventral SNpc DA neurons depend more on T-type calcium channels than dorsal SNpc DA neurons [106,107]. Dependence of calcium channels might contribute to the higher calcium concentration in vSNpc DA neurons and therefore contribute to selective vulnerability.

ALDH1A1: Another marker for selective vulnerability is aldehyde dehydrogenase 1 (ALDH1A1). ALDH1A1+ dopamine neurons express mostly in the ventral tier of the substantia nigra, and therefore serve as a marker separating dorsal and ventral SNpcs [110,117]. This group of dopaminergic neurons preferentially project to dorsal and lateral parts of the rostral and intermediate striatum, and less so to the caudal parts of the striatum [110,118,119,120]. Especially in the DLS, ALDH1A1+ dopaminergic neuron projections are enriched in striosomal DLS regions [110,119]. According to a previous paper, this part of the striatum receives massive cortical projections from the primary and secondary motor cortexes [33]. Thus, this is consistent with the fact that, in Parkinson’s disease, ventral SNpcs with more ALDH1A1 positive neurons degenerate more severely and movement is more affected. Further molecular evidence was shown in experiments conducted on ALDH1A1-deletion animals which suffer from motor skill learning impairments [120]. This is consistent with our discussed motor or habit roles for the DLS in comparison to the more cognitively skewed DMS [9,36,51,67,82,121]. However, it is important to consider the non-motor cognitive and reward learning aspects of the DLS that may also underlie some PD symptomatology [21,52,122,123]. Nevertheless, the selective vulnerability of DLS-projecting DA neurons in comparison to the DMS-projecting DA neurons is broadly consistent with the DMS/DLS dichotomy.

DCC: Having discussed physiological and metabolic molecular factors, it is important to note that recent work has highlighted the role of developmental cascades in PD. For example, deleted in colorectal cancer (DCC) is a cancer/developmental gene that has received attention for its role in PD. DCC was first characterized in colorectal cancers in 1990 [124]. DCC was later discovered to be a ligand of Netrin-1 in the nervous system, mediating axon guidance and neurodevelopment [125,126]. Interestingly, in the rodent brain, DCC has been found to have preferential expression in the ventral tier of SNpc DA neurons compared to dorsal SNpc and VTA [98,114]. This finding is supported by colocalization studies of calbindin and DCC in substantia nigra, which revealed that DCC was rarely colocalized with calbindin in SNpc [127]. Related to our above motor circuit hypothesis, DCC is enriched DA neurons that primarily project to the DLS. Thus, DCC’s expression pattern is consistent with the vulnerability selectivity in PD DA neuron degeneration.

Although we know that Netrin-1/DCC is still expressed in adult SNpcs, the mechanism of DCC-mediated selective vulnerability has been elusive until recently. Using loss-of-function rodent experiments, it was found that Netrin-1 deletion leads to α-synuclein aggregation and that unbound DCC could trigger the cell death pathways [128]. In this work, the authors also showed that overexpression of Netrin-1 could be neuroprotective in PD models. These facts indicated that loss of Netrin-1 and artificially unbound DCC lead to vulnerability in ventral SNpc in an α-synuclein-dependent manner. Thus, with the knowledge of DCC expression in the ventral tier of the SNpc, this work serves to provide molecular mechanistic evidence for the circuit-based motor deficits seen in PD.

PD-related genetic expression: As we discussed above, different expression levels of calbindin, ALDH1A1, and DCC in the dorsal and ventral tiers of SNpc DA neurons and VTA either directly or indirectly contribute to selective vulnerability in DA neurons. Besides these major genes, transcriptomic analyses have been performed to investigate the different gene expression profiles among VTA, dSNpc, and vSNpc. It has been long known that neuroprotective genes are more expressed in VTA and metabolic genes are more expressed in SNpc, which indicates a higher metabolic burden in SNpc [98]. Recent studies have started to dive into subpopulations in the SNpc to further establish the gene expression differences in human tissue. Consistent with what has been discussed, relatively more neuroprotective genes are expressed in the VTA and dorsal tier of SNpc, while more PD-related genes are expressed in the ventral tier of SNpc [129]. Taken together, these studies explain the molecular culprits underlying selectively vulnerability of ventral SNpc, although more studies need to be performed in rodent models for evidence. Here, we invoke the hypothesis that the motor biased symptoms, together with cognitive symptoms, are related to this selective degeneration of the SNpc ventral tier, and thus a preferential loss of DLS-projecting SNpc neurons.

### 3.3. VTA and Disease

As we have seen so far, the VTA is one of the major areas of the brain that embodies dopaminergic neurons, with projections to the ventral striatum and hippocampus. More importantly, the VTA is the origin of the reward pathway where dopamine is released to allow hedonic sensations while also playing a crucial role in memory and learning. Having such important functions in the brain, it is reasonable to assume that there will be diseases related to the dysfunction of VTA DA neurons. Sure enough, addiction, Alzheimer’s disease, eating disorders, depression, learning deficits, schizophrenia, ADHD, and many more have been linked to abnormalities in the VTA and its dopaminergic system [130,131,132]. In this section, however, we will focus on reviewing the recent literature on drug addiction and Alzheimer’s disease in relation to the VTA DA neurons.

#### 3.3.1. Drug Addiction and VTA

Drug addiction has been defined as a chronic and relapsing neuropsychiatric disorder that manifests as compulsive drug-seeking behavior. In addition to the consistent problem with alcohol and cigarette smoking, the opioid epidemic in the United States caused more than 63,000 deaths in 2016 alone [133]. Thus, much time and effort has been attributed to researching the mechanism and circuitry of drug addiction. There has been much advancement in our understanding of drug addiction, providing us with numerous therapeutics to combat it [134]. In this section, we will focus on the VTA-NAc pathway and its function in drug addiction.

The dopamine theory of addiction made major breakthroughs in the late 1900s, with studies showing evidence that brain areas with DA neurons displayed positive reinforcement [135]. Recurrent electrical stimulation in these midbrain regions lead to positive reinforcement and, furthermore, blocking dopamine receptors in these areas prevents the reinforcing effects of psychostimulants. Furthermore, one group suggested that not only stimulants but a broad range of drugs (i.e., opiates, central depressants, cholinergic agonists) can increase extracellular dopamine through different mechanisms [136]. Therefore, dopamine neurons have been the focus of addiction research for the past 60 years. The main source of dopamine is from the DA neurons of the midbrain, including the VTA, which was found to mediate the rewarding effects of several drugs. Animals were able to learn to self-administer drugs into the VTA (nicotine: [137,138], opiates: [139,140,141,142], cocaine: [143,144,145], ethanol: [146,147]), and this effect was diminished with the addition of dopamine antagonists or with lesions to the VTA DA neurons [137,138,148,149]. Moreover, this seems to be specific to the VTA and the regions it projects, as the SNpc has almost no release of dopamine in response to intravenous drug administration or abstinence [150,151]. This is thus broadly consistent with the hypothesis that circuit-specific DA neuromodulation underlies addiction symptomatology.

How do addictive substances impact neural circuits of the VTA? In the early 2000s, an interesting hypothesis was proposed where it was argued that neuronal alterations could occur in the mesolimbic pathway following drug exposure, leading to long-term behavioral changes [152]. This stemmed from observations that DA neurons of the VTA were able to express NMDAR-dependent long-term potentiation (LTP) and long-term depression (LTD) at their excitatory synapses [153]. Subsequent studies were able to show that drugs including cocaine, amphetamine, morphine, ethanol, and nicotine induced LTP on the excitatory synapses of VTA DA neurons [153,154,155,156,157]. This suggested that the synaptic adaptation occurring on the VTA DA neurons played a role in drug experience. However, it was noted that repeated exposure to cocaine did not show further alterations in the synaptic plasticity of VTA DA neurons [158]. Having seen these changes, the next questions to ask were “What mediates this drug-induced LTP in the VTA?” and “Is there a behavioral change due to the drug-induced synaptic plasticity?”. The trafficking of AMPA receptors with specific subunits has been observed to mediate LTP in different regions of the brain [159,160,161,162], but it was not known whether drug-induced LTP followed the same mechanism. With further investigation, it was found that GluR1-containing AMPARs contributed to drug-induced synaptic activity [163] and resulted in behavioral deficits [164,165]. When a GluR1 knockout mouse model was used, the administration of cocaine did not induce the LTP that was observed previously, and the experimentally-induced conditioned place preference was absent [165]. On the other hand, overexpressing this subunit in the VTA resulted in sensitized behavior, a process where repeated administration of stimulants leads to a gradual increase in motor-stimulant response, to morphine [165]. Putting these together, it is clear that exposure to drugs causes alterations at the synapses of VTA DA neurons. However, the lack of additional change with subsequent drug exposure suggests that the VTA plasticity and activity is more involved in the early stages of addiction, contributing to the initial reward stimulus of the drug experience [154,166].

One of the areas that the VTA DA neurons transmits information is the nucleus accumbens (NAc), which is also part of the reward circuit. With the drug-induced synaptic plasticity in DA neurons of the VTA, one could assume that drug exposure would affect the NAc. Indeed, the NAc has also been implicated for addiction, as the depletion of dopamine in the NAc with 6-hydroxydopamine (6-OHDA) led to diminished rewarding effects of cocaine and amphetamine [149,167,168] while increasing the intake of morphine. Moreover, application of amphetamine or DA receptor agonists to the NAc induced conditioned place preference [169,170], which was attenuated by DA receptor antagonists. The NAc is comprised largely of two anatomical parts, the NAc core and the NAc shell. Although there is evidence that both play a role in addiction, the NAc shell has been identified to be more crucial in terms of the rewarding effects of drugs, while the NAc core has been shown to be more involved with drug-seeking behavior. Animals learned to self-administer stimulants or dopamine receptor antagonists into the NAc shell but not the core. Moreover, injections of dopaminergic antagonists into the shell, but not the core, led to decreased rewarding effects induced by drugs [171]. From the pharmacological manipulations, we can see that the dopaminergic system is again mediating these behavioral outcomes.

The differences between the NAc shell and core in addiction may potentially be explained by the different genetic expression of DA neurons in the VTA. One study tried to map the different dopamine subtypes and its connectivity, and it was found that Sox6+ DA neurons of the dorsolateral VTA projected to the NAc core and the lateral shell, while vesicular glutamate transporter 2 (Vglut2+) and aldehyde dehydrogenase 1 family member A1 (ALDH1a1+) DA neurons of the VTA projected to the NAc medial shell and olfactory tubercle (OT) [110]. In addition, cholecystokinin (CCK+) DA neurons were found to project to all regions of the NAc and OT [110]. Interestingly, VGlut2+ DA neurons have been identified to be essential for psychostimulant-mediated behavior activation [172]. ALDH1a1 expression in DA neurons was shown to mediate GABA co-release, and decreased ALDH1a1 expression led to increased alcohol consumption and preference [173]. Although it seems that there are differential projections of VTA DA neuron subtypes, more investigation is needed into what role each DA neuron subtype may have in mediating addiction-related behaviors. Furthermore, it will be informative to link these genetic studies to the input–output relationships seen in the VTA [174], and to determine whether the substance-induced plasticity in DA neurons may have a molecular identity or cause.

The VTA DA neurons are known to project onto GABAergic medium spiny neurons (MSNs), which are the NAc’s major cell type. Interestingly, it has been suggested that there are also alterations in the excitatory synaptic inputs, specifically excitatory cortical afferents, on the MSNs of the NAc shell. Similar to what we saw with the VTA, drug administration leads to altered synaptic strength. Application of cocaine induces long-term depression (LTD) at excitatory synapses found on the GABAergic medium spiny cells, and the absence of this LTD prevented the expression of behavioral sensitization, the process of progressively increasing behavioral response due to repeated administration of stimulants [175,176]. However, in contrast to the VTA, cocaine induced this depression following a period of chronic administration rather than a single dose. This suggests that the NAc is more related to the long-lasting behavioral responses due to previous or repeated drug exposure. To examine the behavioral aspects of this synaptic plasticity in the NAc, GluR1, which facilitates LTP, as mentioned before, was examined. Overexpression of GluR1 in the NAc was found to diminish multiple aspects of drug-associated behavior (drug-seeking behavior and rewarding effect) [177,178], while overexpression of a pore-dead GluR1 led to diminished synaptic strength, amplifying the drug-associated behaviors [179]. Moreover, the role of the NAc shell was emphasized in these studies, as the NAc core was not shown to have alterations in GluR1 subunit AMPA receptors in response to chronic cocaine self-administration [177], again suggesting that the NAc shell has a more prominent role in addiction. In addition, it was recently discovered that the NAc medial shell provides inhibitory control to mesolimbic DA neurons through GABA inhibition [180]. Depression of the excitatory synapses on the GABAergic medial spiny neurons may decrease the activity of this pathway, leading to the overactivity of DA neurons and potentiating the rewarding effects of dopamine. Together, these data suggest that the dopamine system of the VTA and NAc play crucial roles in addiction, with the VTA focused on the early stages of disease development and the NAc responsible for behaviors further along.

Putting all this information together, it is convincing that the VTA-NAc pathway is important and plays a role in understanding drug addiction. Moreover, as the NAc shell receives more input from the medial VTA, one may infer that the medial VTA to NAc shell circuit is more involved in the process of initial drug acquisition. On the other hand, the lateral VTA projecting to the NAc core has been demonstrated to have a role in ongoing motivation [71]. In addition, not only is the VTA compartmentalized by its medial-lateral axis, but it also has differences along its anterior-posterior axis, especially in terms of its projections. Some studies have suggested that the posterior VTA is more involved with opioid drug processing [181]. However, the synaptic changes or behavioral changes depending on the different regions of VTA have not been explored extensively. With more studies elucidating these differences, we will be able to form therapeutics that target certain cell types or regions to combat each individual drug specifically.

#### 3.3.2. Alzheimer’s Disease and VTA

As the average age of our population grows, there is an increasing burden on our healthcare system. Dementia is one of the most prevalent diseases in the aged population, with Alzheimer’s disease (AD) being the primary culprit. AD is a devastating neurological disease characterized by progressive memory loss and brain atrophy. The pathology of this disease has continuously been attributed to amyloid plaques and neurofibrillary tangles, causing massive destruction including neuronal loss and synaptic failure. Therefore, we have constantly been focused on the hippocampus and cortex, which are the areas directly related to the symptoms. However, the extrinsic connections to the hippocampus are commonly ignored when discussing the hippocampus, even though they have been identified to be disrupted in AD [182]. The hippocampus receives sub-cortical inputs from various regions, and specifically receives dopamine from the VTA [183,184,185] and locus coeruleus (LC) [75,76] through tyrosine hydroxylase positive (TH+) fibers. In this section, we will be focusing on the recent literature regarding the VTA-hippocampus-NAc pathway in relation to AD.

Dopamine is known to be crucial for modulating synaptic plasticity and memory encoding in the hippocampus [76,183,184,185,186]. As VTA is one of the major sources of dopamine, one can hypothesize that alterations of the VTA DA neurons may lead to the memory deficits we observe in AD. Sure enough, there have been reports that the DAergic system is altered in AD patients, and it is characterized by decreased levels of dopamine and dopamine receptors [187,188]. Moreover, a recent study provided evidence that there was selective neuronal loss in the VTA causing degeneration of dopamine neurons, even before the formation of amyloid plaques in an AD mouse model [189]. Interestingly, other areas including the LC, the hippocampus, and cortical regions were not affected. To further support these findings, a recent clinical study in humans showed that there was a strong association with VTA size to both hippocampal size and memory index in healthy subjects [190]. In addition, hippocampal volume and memory performance, markers for AD, were found to be associated with the functional connectivity between the VTA and hippocampus. Similar to what we saw in mice, other regions did not share this relationship. The authors who discovered the VTA DA neuronal loss in the AD mouse model also reported that the dopamine outflow to the NAc is decreased [189]. As we have seen, the NAc is also a region that the VTA DA neurons project to, modulating reward and motivation processing. This suggests that some non-cognitive function deficits found in AD, such as behavioral deficits or neuropsychiatric symptoms, may be partially due to alterations of the VTA-NAc pathway. One recent study showed that chemogenetic and optogenetic stimulation in the VTA of a dopamine transporter (DAT)-Cre transgenic mice caused hyperactive locomotion and induced reinforcement, while inhibition or pharmacological interventions (DA receptor antagonist) showed opposite effects [191], supporting this hypothesis.

The hippocampus projects glutamatergic transmissions to the NAc core, thus completing a VTA-hippocampal-NAc-VTA loop [192,193]. The NAc core is known to have functions related to memory, learning [194], and reward behavior [195]. Therefore, it was questioned whether alterations in the DA outflow to the hippocampus can cause changes in the hippocampus-NAc core connection. Indeed, the same AD mouse model above was found to show impaired glutamatergic transmission from the hippocampus to the NAc core (reduced excitability and impaired LTP, which was attributed to the loss of VTA DA neurons [196]). Moreover, the administration of L-DOPA rescued the mouse from all deficits. Putting these together, the loss of VTA DA neurons seems to occur in the preclinical stages of AD, causing deficits in the VTA-hippocampus-NAc core functional pathway, thus causing symptoms of AD.

The studies regarding VTA DA neurons and AD show promising results, especially as a marker or early indicator of AD, as these deficits are found mainly in the preclinical stages of AD. To achieve this goal, outlining the specific anatomical relationships is crucial to understanding this pathway. Only a small portion (6–18%) of the projections from the VTA to the hippocampus are from DA neurons [17,197]. Moreover, these DA neurons predominantly project almost exclusively to the CA1 region of the hippocampus, while the NAc core receives its glutamatergic input from the ventral hippocampus [198,199]. These specific regional relationships suggest a specialized role for these projections. Nevertheless, to understand the full extent of how dysfunction of the DAergic system affects AD, much more investigation of the VTA-hippocampus-NAc pathway is needed.

### 3.4. Schizophrenia, Perception, and Striatal Dopamine

As discussed, major bodies of evidence point toward both reward and motor accounts of DA’s contributions to behavior. However, in each section, we also discussed how striatal circuitry and the dopaminergic modulation thereof may regulate multiple modalities of sensory perception [38,39,53,55,56,57,58,60,200,201,202,203]. Like the reward and motor deficits seen in Parkinson’s disease and addiction, previous and emerging evidence points towards a role for DA dysregulation in clinical perceptual deficits [204,205]. Chief in this regard, schizophrenia and other psychotic disorders are majorly characterized by perceptual disturbances and hallucinations in multiple sensory modalities. Related to DA, DA antagonism has long served as treatment that curtails hallucinations and other positive symptoms seen in schizophrenia. This pharmacological and other neuroimaging evidence supports the dopaminergic hypothesis of schizophrenia, that excessive dopamine underlies schizophrenia and perceptual symptomatology [206,207,208,209]. In addition, in terms of negative symptomatology, decreased phasic DA responses have been thought to underlie decreased levels of learning, attention, motivation, and memory in schizophrenic patients [210]. Consistent with this, DA agonists, amphetamine, and L-DOPA have been reported to induce hallucinations clinically [211]. In mechanistic terms, the system’s relationship between human perception and hallucinations is a topic of heavy investigation [212,213,214]. Current influential theories of how the brain encodes hallucinations is based upon “priors” or a set of perceptual expectations and predictions that the brain makes about the environment [215]. This draws upon the idea that the brain does not encode percepts passively but actively with the engagement of subcortical and memory circuits. In this framework, subjects that develop hallucinatory tendencies may have unusually strong priors or engage in circuits in a manner that emphasizes prior sensory predictions in the absence of real stimuli [204,215].

How do neural circuits (striatal or not) regulate sensory perception and hallucinatory events? Landmark studies of perception and basic choice discrimination support the hypothesis that sensory pathways indeed play a causal role in active perception [216,217,218]. In many of these early studies, cortical microstimulations and microlesions have demonstrated a causal role for sensory pathways in generating perception. Advances in optogenetics and deep brain recordings have paved the way to mechanistically study perceptual decision making and perceptual reporting in deep brain structures, such as the striatum, of animal models. As mentioned previously, optogenetic manipulation of auditory corticostriatal, thamlamostriatal, and striatal neurons has an impact on mouse auditory discrimination [56,57,201]. Similar approaches have provided evidence for striatal perception in other modalities such as somatosensation and vision [38,39,202,203]. While these experiments do not point towards mechanisms of hallucination per se, they invoke the argument that striatal circuits can causally mediate forms of perception.

In a technically and conceptually innovative study, Schmack et al. modeled hallucination-like perception in humans and mice [219]. In their study, they showed that normal humans and mice can report hallucinatory events in an alternative-forced choice task. Briefly, wild-type mice and humans were trained to report the detection of acoustic stimuli and were presented with a range of volumes, from a non-detectable 0 dB tone to 30 dB. Both subject pools reported a small but significant number of choices that corresponded to hearing a sound when none could actually be detected. This paradigm is consistent with previous research supporting strong prior models and detailing such halluncinaton-like events in the absence of neuropsychiatric disease [215,220]. Their systematic behavioral paradigm was validated by the use of hallucinogenic agents, which induced an increased report of false alarms (i.e., hallucination-like events). Interestingly, they reported that an increase in both ventral and tail (auditory) striatal DA preceded trials that resulted in a false report. Furthermore, they demonstrated that optogenetically increasing DA levels in the tail striatum causally increases false alarm report rates. This remarkable and thought-provoking evidence for the DA hypothesis of schizophrenia triggers further questions about the auditory striatum. Further work detailing the nature of auditory corticostriatal activity, MSN stimulus representation, and a characterization of different sensory modalities for hallucinatory-like events will be informative.

### 3.5. Dopamine System Genes and Neuropsychiatric Disease

We have, thus far, discussed the role of DA in various diseases from a systems and neural circuit perspective. However, as further evidence, DA-related genes are implicated as molecular modulators or risk factors for neuropsychiatric diseases such as schizophrenia and addiction [221,222,223,224]. Many of these implicated genes are involved in DA recycling and thus DA homeostasis. Genetic polymorphisms of a dopamine transporter (DAT), a primary enzyme responsible for recycling DA, impact treatment resistance susceptibility and working memory function in patients with schizophrenia [222,225,226]. Furthermore, DAT polymorphisms have been associated with behavioral deficits in anxiety and attention [227,228]. Consistent with this, animal models of DAT genetic manipulation or knockout have shown deficits in executive and cortical function [229]. While DAT levels are high and responsible for recycling dynamics within the striatum, in cortical regions, DAT levels are low and, therefore, DA levels are thought to depend more on degradation dynamics. The catechol-O-methyltransferase (COMT) enzyme is responsible for DA dynamics, and polymorphisms of the COMT gene are also associated with treatment response, cortical activity, and cognitive functioning in schizophrenia [230,231]. Related to this, the relatively common COMT Val^158^Met polymorphism has been associated with differences in behavioral traits such as reward processing and anxiety [232,233]. Together, these genetic studies implicate DA metabolism machinery and thus region-specific DA levels in the expression of behavior in both clinically normal and neuropsychiatric contexts.

## 4. Overall Conclusions/Summary

Since the discovery of DA as a functional neurotransmitter in the 1960s, its role in behavior and disease in the CNS has been heavily investigated. With the advent of genetic, physiological, and systems neuroscience techniques, the fields have begun to draw major conclusions regarding DA. Firstly, DA strongly modulates a diverse range of neural systems, such as the repertoire of the basal ganglia, the cortex and subcortex, and the hippocampus. Through the vast portfolio of the basal ganglia, many behavioral functions arise, such as locomotion, reward processing, and learning. We show that there is a gradient of function across the striatum, with the VS more related to reward/value learning and the DLS more related to sensorimotor habit execution. DA’s specific role in these regions seems to be one of functional support, but there are still open-ended questions to answer. For instance, the clearly distinct roles for DA transmission in each subregion are not quite clear, with auditory striatal DA appearing to be the most distinct. Nevertheless, different diseases have implicated different DA pathways, further pointing toward subregional roles for DA. In rodent and human models of Parkinson’s, the SNpc-DLS pathway seems to be particularly vulnerable, famously sparing the VTA DA neurons. In addition, the role of the VTA in reward processing becomes maladaptive. Additionally, recent evidence indicates that the VTA-hippocampal pathways may be selectively vulnerable to insults from experimentally introduced amyloid. With these exciting advances, there are still many questions that remain open-ended. In the reviewed studies and in future studies, new circuit level technologies such as the use of DA sensors and in vivo imaging will allow us to dissect the contributions of circuits to disease [234,235]. What behaviors do specific dopamine receptor pathways (D1-D5) support and how are they related to DA loss in disease? How does the D1/D2 canonical striatal pathways differ in function in each striatal subdivision? Can subtype specific dopamine agents be fruitful for different neuropsychiatric diseases? Are there major metabolic or genetic principles that underlie the SNpc’s/VTA’s ability to cope with aging or inflammation? These many questions are bound to be asked and addressed as the field develops further understanding through newly introduced technologies.

## Figures and Tables

**Figure 1 biomedicines-09-00647-f001:**
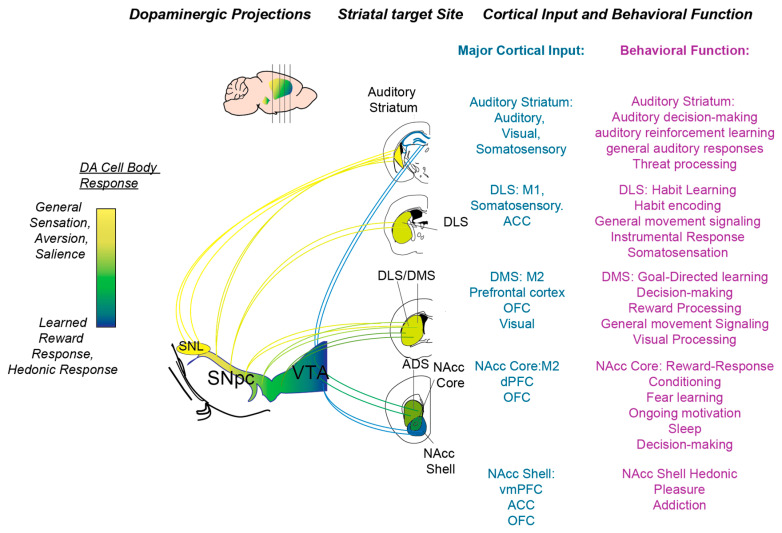
The medial–lateral midbrain projection system to the anterior–posterior striatal axis: Different regions of the dorsal striatum receive dopaminergic inputs from distinct but overlapping populations of the SNpc and the VTA. The posterior auditory striatum primarily receives nigrostriatal input from the lateral portions of the SNpc or what has been termed the substantia nigra pars lateralis (SNL). Moving anteriorly, there is relatively more contribution from the medial portions of the SNpc, and then subsequently contributions from the VTA. On the opposite end of the spectrum, the NAcc ventral striatum receives input primarily from the VTA, with lateral portions of the VTA projecting to the NAcc core and medial portions projecting to the shell. Major input across the posterior to anterior dorsal striatum/NAcc gradually shifts from sensory to limbic cortices. Here, we listed the differential inputs and associated behavioral functions of each subregion. Abbreviations: Dopamine, DA; dorsolateral striatum, DLS; dorsomedial striatum, DMS; nucleus accumbens, NAcc; primary motor cortex, M1; anterior cingulate cortex, ACC; orbitofrontal cortex, OFC; secondary motor cortex, M2; dorsal prefrontal cortex, dPFC; ventromedial prefrontal cortex, vmPFC.

**Figure 2 biomedicines-09-00647-f002:**
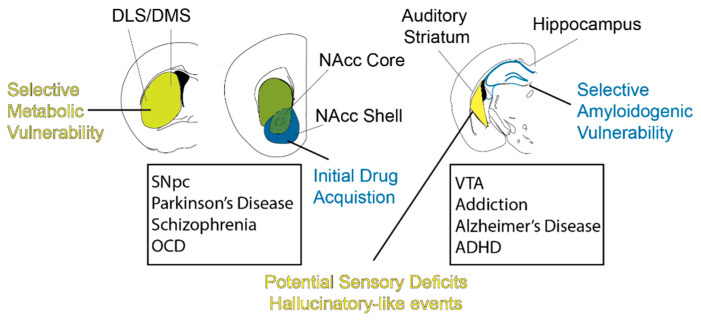
Summary of reviewed DA-related diseases categorized according to different brain regions. Diseases associated with the SNpc include, but are not limited to, Parkison’s, schizophrenia, and OCD. The DLS is subject to metabolic and molecular vulnerability in Parkinson’s disease. The tail striatum has recently been implicated in hallucinatory-like percepts in schizophrenia models. Diseases associated with the VTA include, but are not limited to, addiction, Alzheimer’s disease, and ADHD. The shell of the nucleus accumbens have specifically been linked to hedonic processing and plasticity in initial drug acquisition. The VTA also projects to the hippocampus and it has been found that these neurons are vulnerable in amyloidogenic mouse models.

**Table 1 biomedicines-09-00647-t001:** Highlighted summary of details characterizing each striatal subregion.

Striatal Subregion	Overall Function	D1R/D2R Function	DA Function
DMS	Goal-directed behavior, visual perception, movement	D1R: Positive reinforcement	Reward, aversive processing, movement
		D2R: Punishment	
DLS	Habit formation, skill refinement, sensorimotor integration, somatosensation, movement	D1R: Habit formation and skill execution	Reward, aversive processing, movement
		D2R: Habit/skill refinement	
VS	Motivation, learning	Reward and aversion opponency	Reward, aversive processing, motivation, movement
Auditory/tail Striatum	Auditory decision making, salience, threat processing, sensory processing	D1R MSNs in driving auditory decisions, D2R MSNs in auditory-induced freezing behavior	Threat learning, sensory processing

## Data Availability

Not Applicable.

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
