# Peer review of "Integrating the Roles of Midbrain Dopamine Circuits in Behavior and Neuropsychiatric Disease"

_biomedicines, 2021, doi:10.3390/biomedicines9060647_

Round 1

Reviewer 1 Report

The article is focused on describing the function of the dopamine system in behavior and various neuropsychiatric disorders. The article is well written and reviews a large amount of valuable information. 

  1. The authors discuss the involvement of D1R and D2R in various processes. However, from the text, it is not clear if authors regard D1R for dopamine D1 receptor or D1-like receptors (D1 and D5) and D2R for dopamine D2 receptor or D2-like receptors (D2, D3, D4). And if it is for D1 and D2 receptors some statement about the involvement of other receptors could be done. 
  2. The authors state that D1 in MSN enhance and D2 in MSN depress activity (part 2.3) but I would also suggest adding a description on the difference in D1R and D2R mechanism of action in the introduction - that will help the readers to better understand described involvement of D1R and D2R in physiology and pathology of the brain. 
  3. in part 3.2 there are several citations formatted as (Lautenschlager, 2018) and not in a numerical way. These references are also missing in the reference list. The authors should check the citations in this part. 

Author Response

  1. The authors discuss the involvement of D1R and D2R in various processes. However, from the text, it is not clear if authors regard D1R for dopamine D1 receptor or D1-like receptors (D1 and D5) and D2R for dopamine D2 receptor or D2-like receptors (D2, D3, D4). And if it is for D1 and D2 receptors some statement about the involvement of other receptors could be done. 

We thank the reviewer for critically pointing out the importance of the D1-like and D2-like receptor families, for both systems/behavioral relevance and completeness. We agree that it is a critical to discuss the nuances in the D1-D5 receptor families, as there is a heavy literature describing specific roles for each receptor in both behavior and disease. In our original manuscript, we’ve only described the literature/experiments detailing D1R and D2R on the basis of genetically defined roles within the striatum. To address this issue:

  • We have clarified in our Anatomical Overview section that we primarily discuss the roles of D1R and D2Rs.
  • Within the same section, we’ve added a small statement on the differences of D1-like and D2-like receptor families and their supported roles within the striatum. We added references that we think will help guide the reader to important studies or reviews on this topic.

  1. The authors state that D1 in MSN enhance and D2 in MSN depress activity (part 2.3) but I would also suggest adding a description on the difference in D1R and D2R mechanism of action in the introduction - that will help the readers to better understand described involvement of D1R and D2R in physiology and pathology of the brain. 

We agree with the reviewer that this sort of description would help inform and draw attention of the readers towards this paradigm. We have added a description with relevant references in the part of the introduction where we first introduce the concept of D1R/D2R.

  1. in part 3.2 there are several citations formatted as (Lautenschlager, 2018) and not in a numerical way. These references are also missing in the reference list. The authors should check the citations in this part. 

We appreciate the careful reading and have fixed these errors.

Reviewer 2 Report

Revision of Manuscript: Biomedicines 1223214

Title: Integrating the roles of midbrain dopamine circuits in behavior and neuropsychiatric disease

Chen et al. is a well-written and compelling literature review on dopamine (DA) as a multifunctional neuromodulator that controls different CNS functions and behaviors, with important implications in neuropsychiatric diseases. In particular, the Authors explore the different roles of DA in brain areas including the basal ganglia’s dorsomedial/dorsolateral striatum, ventral striatum, auditory striatum, and the hippocampus in rodent models. Then they highlight that DA dysregulation states in Parkinson’s disease, addiction, and Alzheimer’s Disease, schizophrenia are linked to different DA pathways mediated by DA1 and DA2 receptors and ion channels. The manuscript addresses the topic proposed and the reference list is well balanced and adequately supports the text. Nevertheless, I have some minor comments that need to be addressed before the publication.

Comments:

-The Manuscript does not mention the physiological systems that recycle the DA such as the dopamine transporter (DAT) that predominates in striatum, and the catechol-O-methyltransferase (COMT) that predominates in cortex. Polymorphisms of such genes are recognized to influence several behavioral processes and have been implicated in the pathogenesis of neuropsychiatric diseases such as schizophrenia and addiction. I suggest the Author to add a section on this.

-In Table 1 I would change “role” with “function” in the titles of each column of the table.

Author Response

-The Manuscript does not mention the physiological systems that recycle the DA such as the dopamine transporter (DAT) that predominates in striatum, and the catechol-O-methyltransferase (COMT) that predominates in cortex. Polymorphisms of such genes are recognized to influence several behavioral processes and have been implicated in the pathogenesis of neuropsychiatric diseases such as schizophrenia and addiction. I suggest the Author to add a section on this.

We thank the reviewer for the great suggestion and bringing to our attention the importance of the function of DA recycling enzymes such as DAT and COMT. With the reviewer’s suggestion and how it ties into our schizophrenia/addiction work, we added a section on this in our disease section. We describe some relevant literature in regard to both schizophrenia and general behavioral traits.

-In Table 1 I would change “role” with “function” in the titles of each column of the table.

We have replaced the words.